# Warning Water Level Determination and its Spatial Distribution in Coastal Areas of China

**Shan Liu[1] • Xianwu Shi[1] • Qiang Liu[1] • Jun Tan[1] • Yuxi Sun[1] • Qingrong Liu[2] • Haoshuang Guo[1]**

[1] National Marine Hazard Mitigation Service, Beijing 100194, China

[2] North China Sea Marine Forecast Center, State Oceanic Administration, Qingdao 266100, China

*Correspondence to*: Xianwu Shi (xianwu.shi@mail.bnu.edu.cn)

**Abstract**:The warning water level is the default water level at which storm surges may occur along a coast and indicates a stage of alert. This level forms the basis for storm-surge forecasting, and prewarning is an important reference for governments and aids in the decision-making process for storm-surge prevention and disaster risk mitigation. The warning water level has four categories (blue, yellow, orange, and red) based on water level observational data. Taking into account the actual defense capability of the shore, we determined the warning water level by comprehensively analyzing factors, including the high water level at the typical return period of each shore section, wave exposure degree and defense capability of storm surge protection facilities, and the shore section's importance level. Here, we proposed a quantitative method for determining the warning water level, and the application of this method was introduced by taking the determination of the warning water level at the shore section of Zhifu District (Yantai City, Shandong Province, China) as an example. We analyzed the spatial distribution characteristics of the warning water levels for 259 shore sections along the coast and revealed their current marine disaster prevention capabilities. Our findings provide a valid direction for determining future warning water levels and a reliable scientific reference for redetermining warning water levels in coastal areas while improving marine disaster prevention and protection capabilities.

**Keywords**:Warning water level; Return period; Spatial distribution; Defense capability;

## 1. Introduction

China is severely affected by storm surges, which have caused huge economic losses and casualties in coastal areas and represent an important factor restricting coastal economic and social development. A statistical report showed that storm surges caused 78.407 billion yuan of direct economic losses and 33 deaths from 2012 to 2021 (including missing person cases) along the coast of China (Ministry of Natural Resources of China, 2021). However, the number of deaths due to storm surges has decreased sharply due to improvements in storm surge warning systems (Shi et al., 2015). The warning water level is the default water level at which storm surges may occur along the coast of protected areas, indicating a stage of alert and the need to implement disaster relief strategies (State Oceanic Administration of China, 2012). Notably, the warning water level is the basis of storm-surge forecasting; it also provides a distinct signal to raise an alert for storm-surge disaster prevention and mitigation.

The warning level of a storm surge is determined based on the highest water level of each tide gauge station affected by the storm surge exceeding the local warning water level. A number of simulation models played an important role in the prewarning of storm surges, including Sea, Lake, and Overland Surges from Hurricanes (SLOSH) in the USA, DELFT3D model in Dutch, and MIKE21 model in Denmark (Konishi, 1995; Lenstra et al., 2019; Lin et al., 2010; Mercado, 1994). Several numerical models have been widely applied across various countries and regions to simulate and forecast storm surges and coastal flood inundation. The National Oceanic and Atmospheric Administration used the SLOSH model to jointly conduct storm surge risk assessment with government agencies and make large-scale National Storm Surge Hazard Maps for the Disaster Management department, insurance companies, and residents(National Oceanic and Atmospheric Administration of USA, 2018). The Royal Netherlands Meteorological Institute categorized the country's coastlines into several parts (according to tidal changes), determined the warning water level, utilized the Dutch continental shelf model to forecast storm surges, and issued alerts according to the warning water level (Herman et al., 2013; Yu et al., 2020). High-precision storm surge numerical models were conducted to investigate the inundation range and water depth distribution of storm surges in Pingyang County (Zhejiang Province, China), as well as in Jinshan District (Shanghai, China) and Huizhou District (Guangdong, China) (Shi et al., 2020a; Shi et al., 2020b; Wang et al., 2021). A 2-D flood inundation model (FloodMap-Inertial) was employed to predict coastal flood inundation of Lingang New City(Shanghai, China), considering 100- and 1000-year coastal flood return periods(Yin et al., 2019). Much of the current work on extreme-coastal-flooding events is based on the classical extreme-value theory (EVT), which identifies the family of distribution functions known as generalized-extreme-value (GEV) distribution as a general model for the distribution of maxima (or minima) extracted from fixed time periods of equal length(Stuart, 2011; Maria et al., 2022; Haixia et al., 2022).In the mid-1990s, the State Oceanic Administration of China determined the warning water level for key ports and shore sections based on observational data from ocean stations (Huang and Chen, 1995), and the created determination criterion was one-dimensional; specifically, it was one value per station. With the rapid development of China's marine economy, the coastline characteristics, development status, population density, and protection facilities in coastal areas have greatly changed. Notably, the warning water level determined at the end of the last century is no longer applicable to current conditions or appropriate for storm surge prevention and mitigation. Therefore, the State Oceanic

Administration of China organized a new round of warning water level assessments in coastal areas in the mid-2010s, and the criteria of water warning levels was divided into four categories (blue, yellow, orange, and red) , spanning 259 shore sections in 11 coastal provinces. This assessment was then issued by the governments of each coastal province (National Marine Hazard Mitigation Service of China, 2018). In order to adapt to the new structure of coastal disaster prevention and mitigation, the newly issued warning water levels were quickly applied towards the early warning and forecasting of storm surges (Fu et al., 2017). The four warning water levels corresponded to the four levels of marine disaster emergency response levels (State Oceanic Administration of China, 2015), which significantly strengthened and supported disaster emergency management at all levels of China's coastal governments.

Here, we describe the technical methods used for warning water level determination and introduce the process and results of this determination in Zhifu District in Yantai City, Shandong Province, China. Through the analysis of spatial distribution characteristics of the warning water levels in 259 shore sections in China, we revealed the current marine disaster prevention capabilities of coastal areas, based on which we propose improvements for future warning water level assessments. Notably, this assessment can provide a scientific reference for promoting the redetermination of warning water levels in China's coastal areas and further improve their marine disaster prevention and protection capabilities.

**2. Material and methods**

**2.1. Data**

This study entailed the processing and use of various types of data: the annual maximum observational water level data from the tide gauge stations, storm surge disaster data, wave run-up data, data of storm surge protection facilities, and the socioeconomic data of shore sections. The coastlines of China were divided into 259 shore sections corresponding to coastal county units. More than 120 tide gauge stations were used in this study. For each shore section, we selected one representative tide gauge station.

In order to ensure the scientific reproducibility of the process we used to determine warning tide levels, the process for selecting the representative tide gauge stations of each shore section were as follows: (1) The number of stations is sufficient to cover the coastal areas from north to south; (2) The station is located near the corresponding shore section, making it representative of the characteristics of the shore section; in terms of the tide, waves, and storm surges exhibited by the shore section; (3) If tide gauge station was absent in a shore section, the tide gauge station closest to the shore section was used; (4) It was ensured that each station had observational water level data for at least 5 years.

Based on the above mentioned procedure, four-color warning water levels of the 259 shore sections were determined through the comprehensive analysis of multiple factors, including, the typical return period value of high-water-level at each shore section, degree of wave exposure, actual defense capability of storm surge protection facilities, and the shore section importance level.

**2.2. Different return periods of high water level calculation method**

Based on the annual maximum observational water level data of the tide gauge stations, the Gumbel model was used as a frequency analysis method to evaluate the return period value of the high water level (HWL) at each station. The Gumbel distribution model is shown in Eq. (1):

$$F(x) = e^{-e^{-\frac{x-\mu}{\beta}}} \tag{1}$$

where x refers to the annual maximum sample sequence of HWL, μ refers to the position parameter, and β refers to the scale parameter. The least squares method was selected to obtain μ and β.

The different return period value of HWL "X" is calculated by Eq. (2):

$$X = \mu - \beta ln\left(-ln\left(1 - \frac{1}{T}\right)\right) \tag{2}$$

The return period "T" is calculated by Eq. (3):

$$T = \frac{1}{1 - F(X)} \tag{3}$$

## 2.3. Calculation method of blue, yellow, orange, and red warning water levels

The warning water level is categorized into four types: blue, yellow, orange, and red, which are described in Table 1. The four warning water levels corresponded to the four levels of storm surge disaster emergency response levels: I, II, III, and IV, which are described in Table 2. Storm surge disaster alerts are divided into four levels: red, orange, yellow, and blue, corresponding to the highest to lowest warning water levels, respectively.

Tab.1 Description of the blue, yellow, orange, and red warning water levels

| Warning water level | Description |
| --- | --- |
| Blue | Refers to the water level at which the marine disaster warning department issues a blue warning for a storm surge. When the water level reaches this default value, the coastal protected areas must enter an alert stage, and precautions must be taken against a storm surge. |
| Yellow | Refers to the water level at which the marine disaster warning department issues a yellow warning for a storm surge. When the water level reaches this default value, mild marine disasters may occur along the coast of the protected areas. |
| Orange | Refers to the water level at which the marine disaster warning department issues an orange warning for a storm surge. When the water level reaches this default value, relatively severe marine disasters may occur along the coast of the protected areas. |
| Red | Refers to the maximum water level at which safe operation can be ensured along the coast of protected areas and for the affiliated projects. It is the water level at which the marine disaster warning department issues a red warning for a storm surge. When the water level reaches this default value, severe marine disasters may occur along the coast of the protected areas. |

Tab.2 Description of the storm surge disaster emergency response level

| Storm surge disaster emergency response level | Description |
| --- | --- |

| | |
|---|---|
| I (particularly major disaster) | Affected by tropical cyclones or extratropical weather systems, it is expected that the high tide level of one or more representative tide gauge stations in the affected area will reach the red warning tide level in the future, a red storm surge warning should be issued, and level I marine disaster emergency response level should be launched. |
| II (major disaster) | Affected by tropical cyclones or extratropical weather systems, it is expected that the high tide level of one or more representative tide gauge stations in the affected area will reach the orange warning tide level in the future, an orange storm surge warning should be issued, and level II marine disaster emergency response level should be launched. |
| III (relatively major disaster) | Affected by tropical cyclones or extratropical weather systems, it is expected that the high tide level of one or more representative tide gauge stations in the affected area will reach the yellow warning tide level in the future, a yellow storm surge warning should be issued, and level III marine disaster emergency response level should be launched. |
| IV (normal disaster) | Affected by tropical cyclones or extratropical weather systems, it is expected that the high tide level of one or more representative tide gauge stations in the affected area will reach the blue warning tide level in the future, a blue storm surge warning should be issued, and level IV marine disaster emergency response level should be launched. |

The blue warning water level was determined based on HWL at the return period of 2 to 5 year of the shore section and the blue warning water level correction value. The calculation method for the blue warning water level ($H_b$) is shown in Eq. (4):

$$H_b = H_s + \Delta h_b, \tag{4}$$

where $H_s$ is the HWL at the return period of 2 to 5 years; $\Delta h_b$ is the blue warning water level correction value. $H_s$ was determined using the actual defense capability of the shore section. Its respective water level return period was the return period corresponding to the elevation of the top of the dike having the lowest defense capability in the shore section. The method to obtain the value is shown in Table 3. $\Delta h_b$ was determined via comprehensive analysis of natural factors including wind, wave, and tide of previous storm surges, along with the actual defense capability and economic conditions of the shore section. The calculation method is shown in Eq. (5):

$$\Delta h_b = h_1 + h_2 + h_3, \tag{5}$$

where $h_1$ is the adjusted value of wave exposure of the surge protection facilities determined by the wave run-up (R) at the return period of 2 years in front of the dike in the shore section. The method to obtain the value of $h_1$ is shown in Table 4 and this value is negative. $h_2$ is the adjusted value of the surge protection facility construction standard, which is determined based on the difference "$\triangle$" between the elevation of the top of the dike and $H_s$. This value is low where "$\triangle$" is low. The method used to obtain the value of $h_2$ is shown in Table 5. $h_3$ is the adjusted value of the shore section importance level, which is determined by the socioeconomic factors of the shore section. This value is low where the shore section importance level is high. The methods used to obtain the value of $h_3$ and classify the shore section importance level are both shown in Table 6.

149

Tab. 3 $H_s$ value corresponding to return period (unit: a)

| Corresponding water level return period of the actual defense capability of the shore section | Corresponding return period of $H_s$ |
|---|---|
| (0,50) | 2 |
| (50,100) | 3 |
| (100,200) | 4 |
| ≥200 | 5 |

151

Tab. 4 $h_1$ value (unit: cm)

| Wave exposure degree | Severe | Relatively Severe | Moderate | Mild |
|---|---|---|---|---|
| Wave run-up occurs once in 2 years (R) | ≥150 | [100,150) | [50,100) | < 50 |
| $h_1$ | −15 %R | [−15 %R,−10 %R) | [−10 %R,−5 %R) | [−5 %R,0) |

Table note: 1)The wave exposure degree of the storm surge protection facilities depends on the degree of wave reception of the embankment, the water depth at the bottom of the embankment and the wave height at the bottom of the embankment.

2)R is the value of the wave run-up occurs once in 2 years. There is a certain correspondence between the wave exposure degree and R.

3)The value of $h_1$ can be taken as 0~15% of the R.

153

Tab. 5 $h_2$ value (unit: cm)

| Breakwater | $\triangle^* ≤1.24$ m; Sand embankment or natural flat coast | $\triangle^* = 1.25$ m to 1.99 m; Half slope stone embankment dike | $\triangle^* = 2.00$ m to 2.99 m; Stone embankment or component revetment dike | $\triangle^*>3.0$ m; Cement dike |
|---|---|---|---|---|
| $h_2$ | [−20, −10) | [−10,0) | [0,10) | [10,20] |

Table note: The defense capability of storm surge protection facilities(breakwater) is closely related to the dike construction standards. "$\triangle$" is the value of the difference between the elevation of the top of the dike and $H_s$. The value of $h_2$ can be taken as -20~20.

155

Tab. 6 $h_3$ value (unit: cm)

| Shore section level | Definition | $h_3$ |
|---|---|---|
| Particularly important | The shore section level is considered to be particularly important if it meets one of the following conditions:<br>—Population density in the protected area ≥1000 persons/km²;<br>—Port throughput ≥ $3 \times 10^{11}$ kg/a;<br>—Construction investment ≥ $1.4 \times 10^9$ USD;<br>—Economic output of the protected area ≥ $7 \times 10^5$ USD/hm²/a;<br>—The cargo unloading capacity of the central fishing port ≥ $8 \times 10^7$ kg/a;<br>—Agricultural reclamation area ≥ $2 \times 10^3$ hm². | [−20, −10) |
| Important | The shore section level is considered to be important if it meets one of the following conditions:<br>—Population density in the protected area = [400 persons/km², 1000 persons/km²);<br>—Port throughput = [$2 \times 10^{11}$ kg/a, $3 \times 10^{11}$ kg/a);<br>—Construction investment = [$0.7 \times 10^9$ USD, $1.4 \times 10^9$ USD);<br>—Economic output of the protected area = [$1.4 \times 10^5$ USD/hm²/a, $7 \times 10^5$ USD/hm²/a);<br>—The cargo unloading capacity of the first-class fishing port ≥ $4 \times 10^7$ kg/a;<br>—Agricultural reclamation area = [$6.67 \times 10^2$ hm², $2 \times 10^3$ hm²). | [−10,0) |

| | | |
|---|---|---|
| Relatively important | The shore section level is considered to be important if it meets one of the following conditions:<br>—Population density in the protected area = [30 persons/km², 400 persons/km²);<br>—Port throughput = [1 × 10¹¹ kg/a, 2 × 10¹¹ kg/a);<br>—Construction investment = [0.14 × 10⁹ USD, 0.7 × 10⁹ USD);<br>—Economic output of the protected area = [0.56 × 10⁵ USD/hm²/a, 1.4 × 10⁵ USD/hm²/a);<br>—The cargo unloading capacity of the second-class fishing port ≥ 2 × 10⁷ kg/a;<br>—Agricultural reclamation area = [67 hm², 667 hm²). | [0,10) |
| Normal | The shore section level is considered to be normal if it meets one of the following conditions:<br>—Population density in the protected area < 30 persons/km²;<br>—Port throughput < 1 × 10¹¹ kg/a;<br>—Construction investment < 0.14 × 10⁹ USD;<br>—Economic output of the protected area < 0.56 × 10⁵ USD/hm²/a;<br>—The third-class fishing port can meet the berthing demand of local fishing boats;<br>—Agricultural reclamation area < 67 hm². | [10,20] |

Table note: The shore section level is categorized into four grades: particularly important, important, relatively important and normal. Each grade is mainly judged from 6 criteria, as long as one of the criteria is met, the shore section importance level can be considered to be this grade. The six criterion are population density, port throughput, construction investment, economic output, cargo unloading capacity and agricultural reclamation area. The value of $h_3$ can be taken as -20~20.

The red warning water level was determined based on the minimum value of HWL at the return period corresponding to the actual defense capability of all dikes in the shore section and the red warning water level correction value. The equation used to determine the red warning water level ($H_r$) is shown below:

$$H_r = H_d + \Delta h_r, \tag{6}$$

where $H_d$ is the minimum value of HWL at the return period corresponding to the actual defense capability of all dikes in the shore section. $\Delta h_r$ is the red warning water level correction value. The calculation method for $\Delta h_r$ is shown in Eq. (5); the values of $h_1$ and $h_3$ were calculated by the same method used to determine $\Delta h_b$. When calculating $h_2$, "$\Delta$" is the difference between the elevation of the top of the dike and $H_d$.

The yellow and orange warning water levels were determined based on interpolation of the blue and red warning water levels, respectively. The calculation methods for the yellow ($H_y$) and orange ($H_o$) warning water levels are shown in Eqs. (7) and (8), respectively:

$$H_y = H_b + (H_r - H_b)/3, \tag{7}$$

$$H_o = H_b + 2(H_r - H_b)/3. \tag{8}$$

## 3. Results

### 3.1. Determination result of warning water level at a representative shore section

For warning water level determination, we selected the shore section of Zhifu District, Yantai City, Shandong Province, China(Fig. 1); the representative tide gauge station for this shore section was the Zhifudao tide gauge station. We considered the annual maximum HWL for 31 consecutive years at the

Zhifu Island tide gauge station and established a frequency distribution curve of the annual HWL using the Gumbel distribution (Fig. 2). The HWL at different return periods obtained using this method are presented in Table 7.

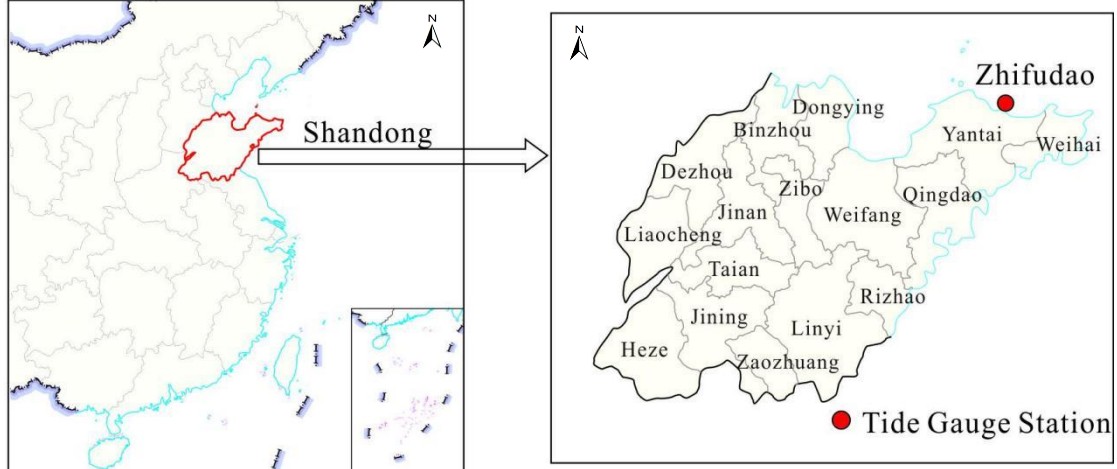

Fig. 1 The location of Zhifudao tide gauge station in Yantai City, Shandong Province, China

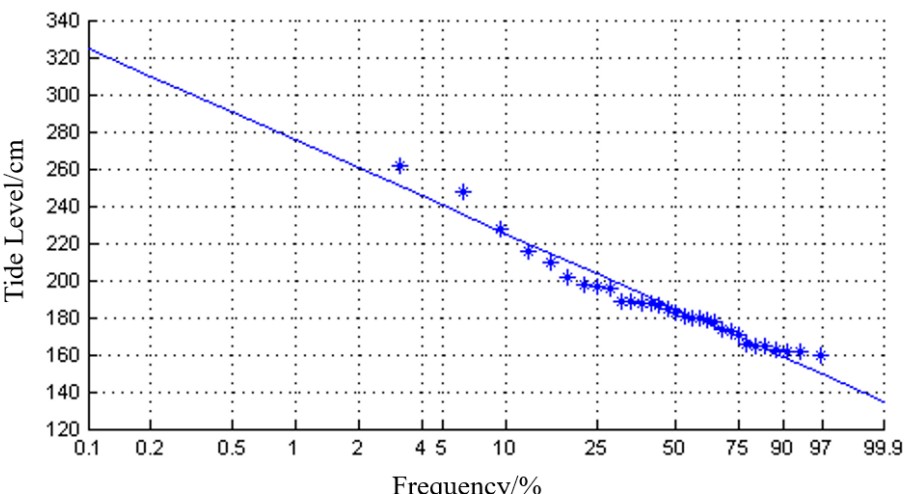

Fig. 2 Frequency distribution of the annual maximum value of the high water level at Zhifudao tide gauge station

Tab. 7 The high water levels (HWL) corresponding to return period at Zhifudao tide gauge station(unit: cm)

| Return period | 2a | 5a | 10a | 20a | 50a | 100a |
|---|---|---|---|---|---|---|
| HWL corresponding to return period | 184 | 209 | 225 | 240 | 260 | 275 |

The actual defense capability of the dike in this shore section corresponded to the return periods of 20 to 50 years. $H_s$ indicated the corresponding HWL at the return period of 2 years, and $H_s$ was 184 cm. The wave run-up that occurs once in two years at the storm surge protection facility in this shore section was 1.0 m. The wave withstand degree was moderate, and $h_1 = -10\% R = -10\% \times 1.0\ \mathrm{m} = -0.10\ \mathrm{m} = -10\ \mathrm{cm}$.

The types of coastal storm surge protection facilities in this shore section included cement dikes, and the
"$\triangle$" for $H_b$ was slightly greater than 3.0 m; therefore, $h_2$ for $H_b$ was 16 cm. The shore section was
considered to be particularly important, thus, the adjusted value of the shore section importance level $h_3$
was valued as −11 cm. The blue warning water level correction value of the shore section $\Delta h_b = -10 + 16$
− 11= −5 cm. The blue warning water level value was calculated to be $H_b = 184 - 5 = 179$ cm.
$H_d$ indicated the corresponding HWL at the return period of 20 years and was 240 cm. For this shore
section, $h_1 = -10$ cm, $h_3 = -11$ cm. The"$\triangle$" for $H_d$ was approximately 2.5 m; therefore, $h_2$ for $H_d$ was 9
cm. The red warning water level correction value for this shore section $\Delta h_r = -10 + 9 - 11 = -12$ cm. The
red warning water level was calculated to be $H_d = 240 - 12 = 228$ cm.
The yellow warning water level was calculated to be $H_y = 179 + (228 - 179)/3 = 195$ cm. The orange
warning water level was calculated to be $H_o = 179 + 2 \times (228 - 179)/3 = 212$ cm.
The warning water level of the shore section in Zhifu District is presented in Table 8.

Tab. 8 Warning water level value of the shore section in Zhifu District, Yantai City, Shandong
Province, China (unit: cm)

| Warning water level | Blue | Yellow | Orange | Red |
|---|---|---|---|---|
| Warning water level value | 179 | 195 | 212 | 228 |


## 3.2. Spatial distribution of warning water level along the coast of China

Using the abovementioned method, the warning water levels of 259 shore sections along the coast of
China were obtained. The spatial distribution maps of warning water level, shore section importance
level, $H_s$, $H_d$, $\Delta h_b$ and $\Delta h_r$ in the coastal areas of China were drafted(Fig. 3; Fig. 4; Fig. 5; Fig. 6).
The warning water level in China's coastal areas was generally low in the northern and southern shore
sections and high in the central shore sections. The maximum warning water levels appeared in the shore
sections in Hangzhou, Zhejiang Province, in the central coastal area of China. The blue, yellow, orange,
and red warning water levels were calculated as 700 cm, 740 cm, 780 cm, and 820 cm, respectively. The
spatial distribution of shore section importance level were consistent with that of the warning water
level. Among the 259 shore sections, the particularly important shore section accounted for the largest
proportion(49.1%), while the other important grades shore sections accounted for 32.4%, 13.1% and
5.4% respectively. The shore section importance levels of Jiangsu, Zhejiang, Fujian, and Guangdong
Provinces were higher than the other shore sections, and more than 90% of the particularly important
shore sections were distributed in the coastal areas of the above provinces. This is because the coastal
zones of these provinces with a high population density were the main areas of economic development
on a country-wide scale, with this importance also being reflected in the high shore section importance
level. The spatial distribution characteristics of $H_s$ and $H_d$ were consistent with that of blue and red
warning water levels, respectively; this can be mainly attributed to the HWL at the typical return period
being the decisive factor in warning water level determination. The warning water level was high where
HWL, at the typical return period, was high. The spatial distribution characteristics of $\Delta h_b$ and $\Delta h_r$ were
similar, but opposite to that of $H_s$ and $H_d$. Figure 6 shows that $\Delta h_b$ and $\Delta h_r$ were generally low in the

central shore sections and high in the northern and southern shore sections. In general, the warning water level correction value $\Delta h_b$ and $\Delta h_r$ was low where shore defensive capability was high.

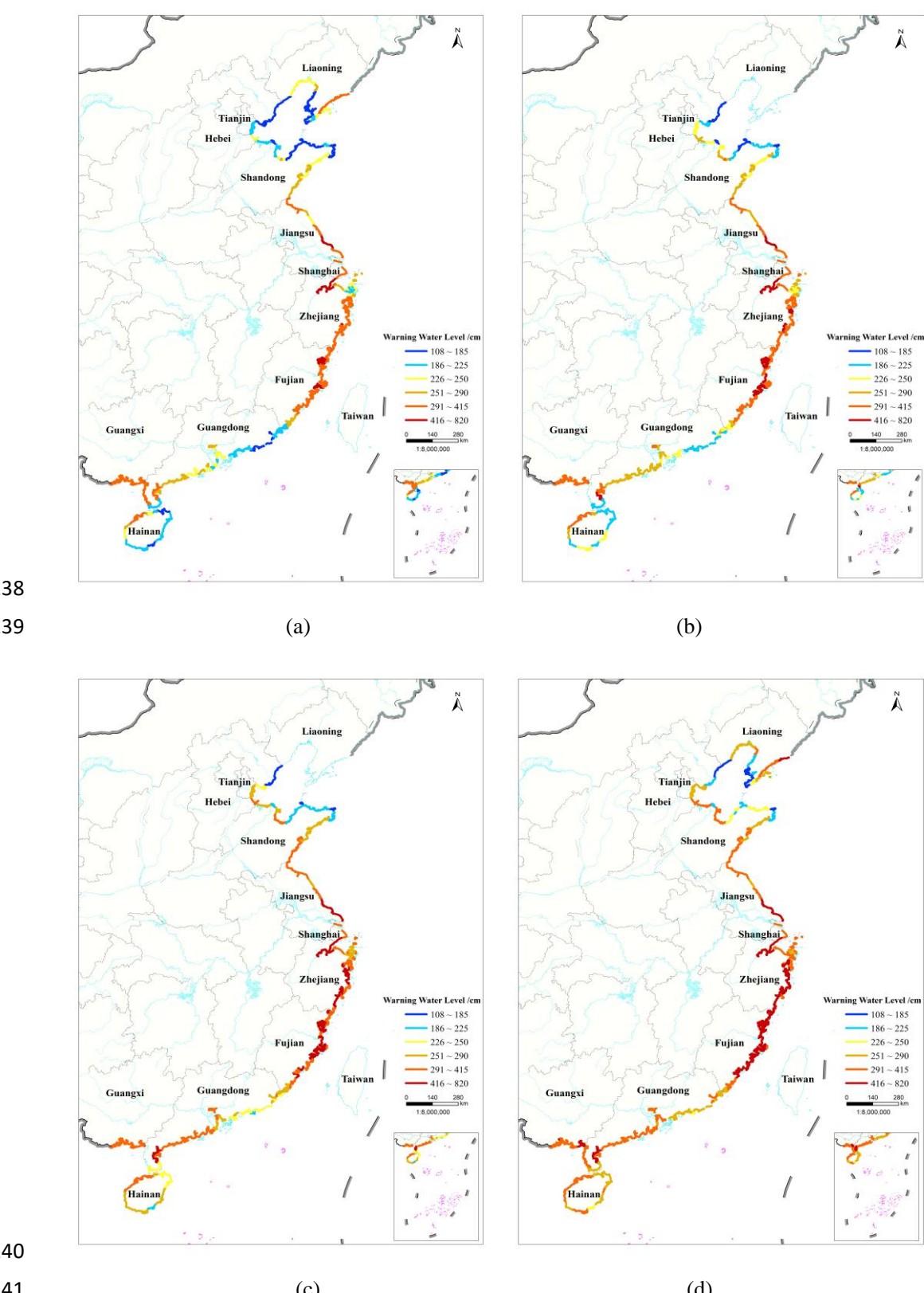

(a)                  (b)

(c)                  (d)

Fig. 3 Spatial distribution map of the four-color warning water level: a) Blue; b) Yellow; c)Orange; d)Red

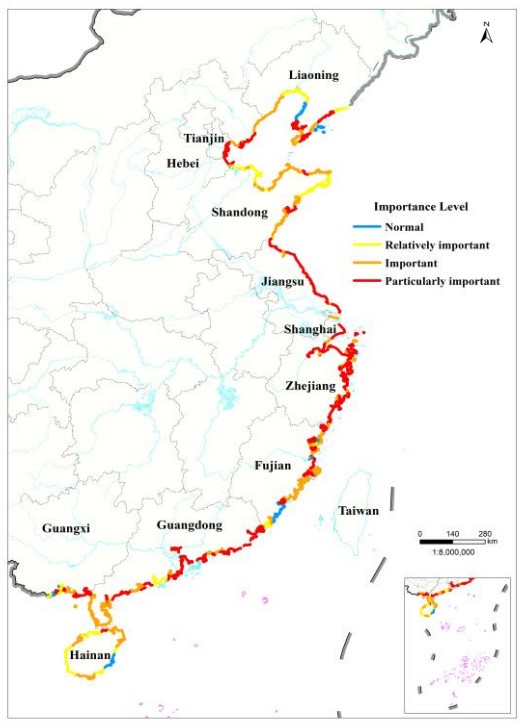


Fig. 4 Spatial distribution map of the shore section importance level

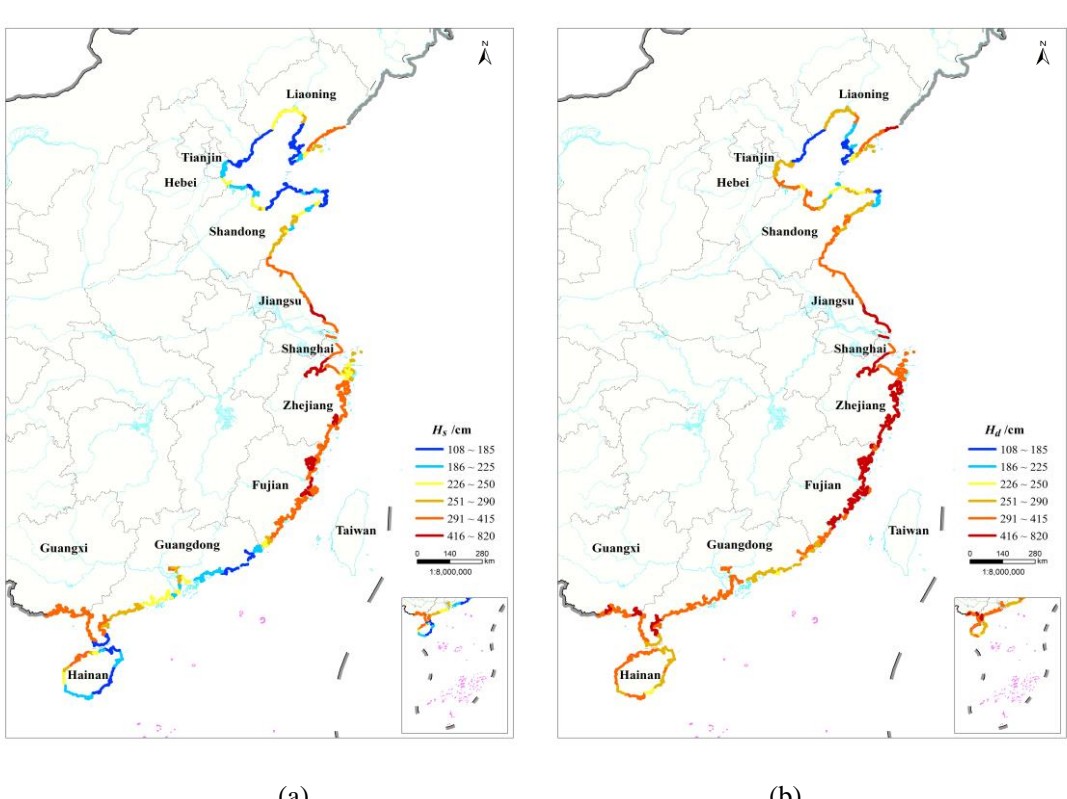


(a)                                    (b)

Fig.5 Spatial distribution map of $H_s$ and $H_d$: a) $H_s$; b) $H_d$





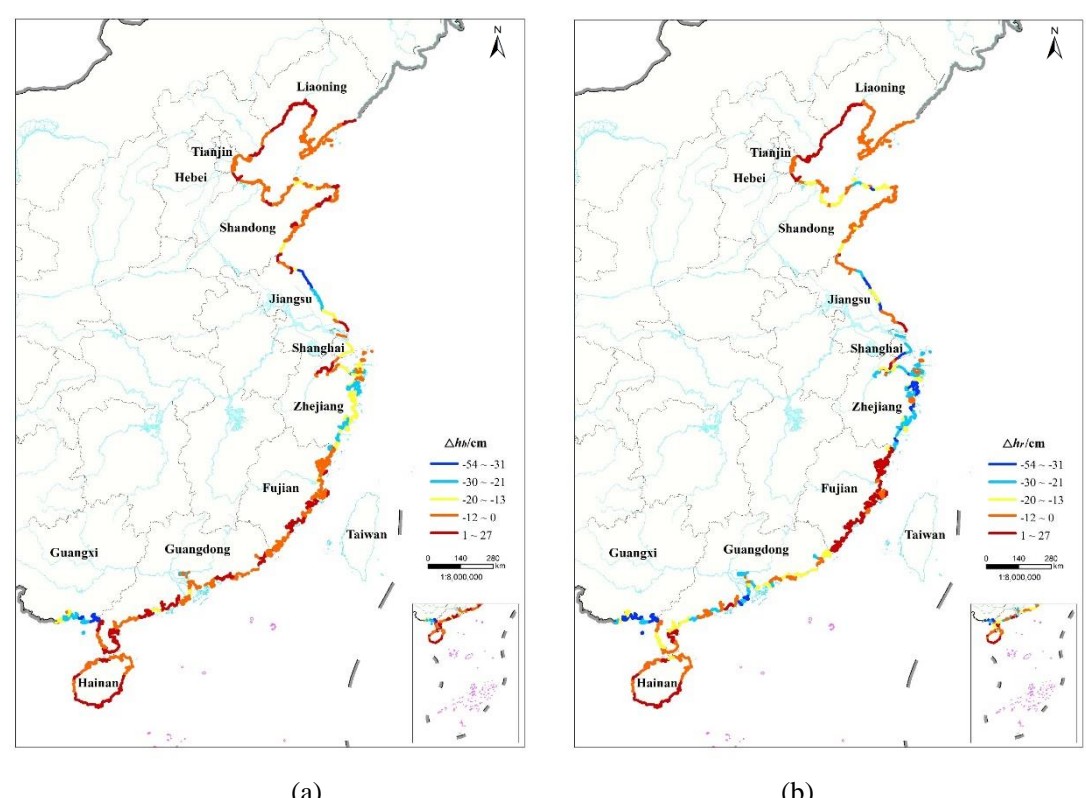

(a)                    (b)

Fig. 6 Spatial distribution map of the warning water level correction value: a)$\Delta h_b$; b) $\Delta h_r$

In the northern coastal areas, including Liaoning, Hebei, Tianjin, and Shandong Provinces, the warning
water level was generally low. These areas are mainly affected by storm surges typical of the temperate
zone, which are of relatively low frequency and intensity. Based on previous observational data, the
calculated water level at the typical return period of the northern coastal areas was lower, indicating the
lower $H_s$, $H_d$ and warning water level.
The shore sections in the central coastal areas, including Shanghai, Zhejiang Province, and Fujian
Province, had higher warning water levels. These areas are mainly affected by typhoon surges of high
frequency and intensity. Moreover, most of the harbors in these provinces are flared or narrow, which can
easily induce larger storm surges, and the water level at the typical return period is greater than that of the
other shore sections, leading to higher $H_s$, $H_d$ and warning water levels in these areas. Notably, the dike
defense capability in these areas is higher, especially for the shore section of Hangzhou Bay in Zhejiang
Province, where the large tidal range leads to an extremely high water level at the typical return period.
Therefore the warning water level in the shore section of Hangzhou Bay is generally higher than that of
other shore sections, indicating the high warning water level distribution in China's coastal areas.
The warning water level in the southern coastal areas, including Guangdong, Guangxi, and Hainan
Provinces, was generally low. Coastal areas in Guangdong and Guangxi Provinces had a lower tidal
range, lower water level at the typical return period, and higher shore section importance level indicating
lower warning water level correction value leading to lower warning water levels. Hainan Island has
more natural coastlines of lower shore defensive capability. This island is less affected by typhoons, and
thus, has a lower high water level at the typical return period, resulting in a lower warning water level.
**4. Discussion**
The warning water level is mainly used for storm surge prewarning, and it is crucial to decision-making
and mitigation measure design. This study proposed a newly approved quantitative method for
determining the four-color warning water level, which includes the calculation formula of the HWL at
the typical return period, the classification method of the shore section based on its importance and
coastal county unit,and the quantitative calculation formula of the correction value of the warning tide
level corresponding to wave exposure degree, surge protection facility construction standard and the
shore section importance level. Compared with the method used for calculating the one-single-value
warning water level in the mid-1990s, the method of calculating the four-color warning water level used
in this study is more reasonable, mainly in the following aspects: (1) It proposed the description of the
warning water level classification corresponding to the four levels of marine disaster emergency
response levels, and the determination results of the four-color warning tide level are more helpful for the
storm surge prewarning, in a way, the newly determined red warning tide level can more truly reflect the
actual defense capability of the shore section; (2) The calculation of correction values has been improved,
by replacing qualitative calculation method with quantitative calculation method, especially proposing
the method of calculating the wave run-up which is an important decisive element for the correction
values; (3) In the process of calculating the four-color warning water level, the verification of the
approved results are strengthened, to determine whether the approved warning water level is suitable
based on the statistical analysis of historical storm surge disasters and the corresponding tidal heights.
Our results about the spatial distribution of four-color warning water level, have been preliminarily
applied to storm surge disaster prevention and mitigation in coastal areas of China. Several studies
focused on the storm surge prewarning application methods for the newly approved four-color warning
water level, corresponding to a refined shore section (Fu et al., 2017). However, limited by the data
availability, it is not considered that the influence of storm surge disaster loss factors on the calculation of
warning water level. The Correlation between storm surge disaster losses and the highest tide water
exceeding the warning water level has not been established.
The precision of the warning water level directly affects the accuracy of the storm surge prewarning
results, thereby affecting the objectivity of emergency strategies and decision-making for storm surge
disaster mitigation. With the rapid development of China's coastal society and economy, storm surge
protection facilities, population density, and coastal development conditions have also been changing.
Therefore, the warning water level needs to be updated according to the actual conditions of the coastal
areas in time, when it is not compatible with the storm surge prevention and mitigation. At the same time,
in order to meet the needs of the increasingly refined storm surge disaster prevention and mitigation plans,
the scale of warning water level assessment should be changed from coastal counties to coastal towns and
communities.
Several studies highlighted that global sea-level rise would continue accelerating in the 21st century as a
consequence of climate change (Church and White, 2011; Hay et al., 2015). In fact, coastal flooding
hazard has been increasing on a global scale in recent decades, a trend expected to continue as a result of
climate change (Maria et al., 2022). In the past 40 years, sea level in the coastal China seas has increased
significantly, with the rate of 3.4 mm/a, higher than the global average from 1993-2018(3.25mm/a)
(Ministry of Natural Resources of China, 2021; IPCC,2021). In the IPCC Sixth Assessment Report, the
latest monitoring and simulation results indicate that the current rate of Global mean sea level rise from

2006 to 2018 is accelerating (3.7mm/a) and will continue to rise in the future, showing an irreversible trend (Zhang et al., 2021; IPCC,2021). Regional relative sea level rise is an important driving factor affecting extreme still water levels. The continuous rising sea level has led to an increase in extreme water levels in coastal areas of China (Qi et al., 2019), which can have an impact on the determination of warning water levels. Additionally, changes in storminess may have an important role in modifying the frequency and magnitude of water level extremes (Lowe et al., 2010; Woodworth et al., 2011). Future work about re-determining the warning water level should take these abovementioned issues into consideration.

**5. Conclusion**

This study proposed an effective method for determining the four-color warning water level, and introduced the application of this method by taking the determination of the warning water level at the shore section of Zhifu District (Yantai City, Shandong Province, China) as an example. Observational water level data from representative tide gauge stations along the 18,000 km coastline were collected and used in this study. Using the method and observational data, we calculated the warning water levels of 259 shore sections along the coast of China and analyzed the assessment results about the spatial distribution characteristics of the blue, yellow, orange, and red warning water levels.

The results showed that the warning water levels were lower in the shore sections of the northern and southern coastal areas in China and higher in central coastal areas. In the northern coastal areas, where are mainly affected by the extratropical storm surges with low intensity, the defense capability of the shore sections was generally low, resulting in the lower warning water levels than the other coastal areas. The maximum values of the blue, yellow, orange, and red warning water levels all appeared in Hangzhou Bay (700 cm, 740 cm, 780 cm, and 820 cm, respectively) of central coastal areas in China. These areas are mainly affected by the typhoon surges with high frequency and intensity, where the defense capability was also high. Understanding the spatial distribution of warning water levels in China's coastal areas cannot only provide important references for national and local governments to aid in the decision-making process for storm surge disaster prevention and mitigation, but also offers a scientific basis for coastal spatial planning, rational layout of coastal industries, and construction of major projects and industrial parks.

**Disclosure statement**

The authors declare that there is no conflict of interest.

**Funding**

This study is funded by the National Natural Science Foundation of China (41701596) and high level scientific and technological innovation talent project of the Ministry of Natural Resources.

**Author Contributions**

Shi Xianwu organized the research project and prepared the manuscript with contributions from all co-authors. Specifically, Liu Shan wrote the manuscript and participated in the calculation of warning water levels; Liu Qiang devised a method for calculating warning water levels; Tan Jun organized the observational data from various tide gauge stations; Sun Yuxi analyzed the distribution of warning water

levels along the coast of China; Liu Qingrong participated in the determination of warning water levels in
the shore section of Zhifu District; Guo Haoshuang participated in designing and drawing the diagrams.
**Data availability statement**
All data used during the study are available from the corresponding author by request.

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
