# Peer review of "Warning Water Level Determination and its Spatial Distribution in"

_Natural Hazards and Earth System Sciences, 2022_

## Author Comment (AC1)

**Responses to Reference Report #1**

This study proposed a method for the determining of warning water level for storm surge and investigated their variations in China, which could provide useful guidance for storm surge early warning and is of great practical significance. While, the manuscript should be more relevant as a scientific article and the scientific contribution of this study should be more clear. Specially,

**Response**: We greatly appreciate your kind help in reviewing the manuscript and all constructive comments. We substantially revised the paper based on these comments.

Comments:

**1. The literature overview in the Introduction is not sufficient; The review of existing studies especially for some latest researches should be strengthened to reveal the research gap and justify the novelty and contribution of this study.**

**Response:** Thanks for your suggestion. More literature on storm surge hazard analysis and return period tide level calculation method have been added in the revised manuscript as follows.

(1) The data of the losses of historical storm disasters in the coastal areas of China has been updated as below:

*Line28-31:*

*A statistical report showed that storm surges caused 78.407 billion yuan of direct economic losses and 33 deaths from 2012 to 2021 (including missing person cases) along the coast of China (Ministry of Natural Resources of China, 2021)*

(2) More detailed explanations about the latest researches are presented in the section of Introduction in the revised manuscript as below:

*Line28-31:*

*The warning level of a storm surge is determined based on the highest water level of each tide gauge station affected by the storm surge exceeding the local warning water level. A number of simulation models played an important role in the prewarning of storm surges, including Sea, Lake, and Overland Surges from Hurricanes (SLOSH) in the USA, DELFT3D model in Dutch, and MIKE21 model in Denmark (Konishi, 1995; Lenstra et al., 2019; Lin et al., 2010; Mercado, 1994). Several numerical models have been widely applied across various countries and regions to simulate and forecast storm surges and coastal flood inundation. The National Oceanic and Atmospheric Administration used the SLOSH model to jointly conduct storm surge risk assessment with government agencies and make large-scale National Storm Surge Hazard Maps for the U.S. Disaster Management department, insurance companies, and residents(National Oceanic and Atmospheric Administration of USA, 2018). The Royal Netherlands Meteorological Institute categorized the country's coastlines into several parts (according to tidal changes), determined the warning water level, utilized the Dutch continental shelf model to forecast storm surges, and issued alerts according to the warning water level (Herman et al., 2013; Yu et al., 2020). High-precision storm surge numerical models were conducted to investigate the inundation range and water depth distribution of storm surges in Pingyang County (Zhejiang Province, China), as*

*well as in Jinshan District (Shanghai, China) and Huizhou District (Guangdong, China) (Shi et al., 2020a; Shi et al., 2020b; Wang et al., 2021). A 2-D flood inundation model (FloodMap-Inertial) was employed to predict coastal flood inundation of Lingang New City(Shanghai, China), considering 100- and 1000-year coastal flood return periods(Yin et al., 2019). Much of the current work on extreme-coastal-flooding events is based on the classical extreme-value theory (EVT), which identifies the family of distribution functions known as generalized-extreme-value (GEV) distribution as a general model for the distribution of maxima (or minima) extracted from fixed time periods of equal length(Stuart, 2011; Maria et al., 2022; Haixia et al., 2022).*

(3) The references added are as below:
*1)Wang, S., Mu, L., Yao, Z.F, Gao, J., Zhao, E.J., and Wang, L.Z.: Assessing and zoning of typhoon storm surge risk with a geographic information system (GIS) technique: a case study of the coastal area of Huizhou, Nat. Hazards Earth Syst. Sci., 21, 439–462, https://doi.org/10.5194/nhess-21-439-2021, 2021.*
*2)Stuart, C. : An introduction to statistical modeling of extreme values, Springer, London, UK, 2011.*
*3)Maria, F.C., Marco, M.: Extreme-coastal-water-level estimation and projection: a comparison of statistical methods, Nat. Hazards Earth Syst. Sci., 22: 1109–1128, 2022.*
*4)Haixia, Z., Meng, C., and Weihua, F.: Joint probability analysis of storm surge and wave caused by tropical cyclone for the estimation of protection standard: a case study on the eastern coast of the Leizhou Peninsula and Hainan Island of China, EGUsphere [preprint], https://doi.org/10.5194/egusphere-2022-847, 2022.*

**2. The methods used in this study should be evaluated, for example, the reason of using Gumbel model for return period analysis should be given and the significance of the fitting should be evaluated; the rationality and effectiveness for the method of calculating the four warning water levels should be clarified.**

**Response:** Thanks for your suggestion. According to your advice, we have summarized the rationality of the methods for return period analysis and four-color warning water levels calculation used in this study. More references have been cited in the revised manuscript. The description of modifications are as follows:

(1) The high tide level at the return period is the extreme tide level value that occurs once in many years. The calculation method is based on the extreme value theory which can be used to simulate the probability and distribution of extreme events (Stuart, 2011; Shi et al., 2011), by establishing the annual extreme value sequence of tide level and selecting an appropriate theoretical frequency curve. The selection of the theoretical frequency curve is based on the principle of the best fitting data. The Gumbel model (Type I extreme value distribution law) is widely used in determining the line type of the tide level frequency analysis at the return period (State Oceanic Administration of China, 2012; Ministry of Transport of China, 2013). In terms of adaptability to the selected high tide level data samples, the Gumbel model has more advantages.

(2) Compared with the method used for calculating the one-single-value warning water level in the mid-1990s, the method of calculating the four-color warning water level used in this study is more reasonable, mainly in the following aspects: 1) It proposed the description of the warning water

level classification corresponding to the four levels of marine disaster emergency response levels, and the determination results of the four-color warning tide level are more helpful for the storm surge prewarning; 2) The calculation of correction values has been improved, by replacing qualitative calculation method with quantitative calculation method, especially proposing the method of calculating the wave run-up which is an important decisive element for the correction values; 3) In the process of calculating the four-color warning water level, the verification of the approved results are strengthened. Statistical analysis of historical storm surge disasters and the corresponding tidal heights is carried out to determine whether the approved warning water level is suitable. If it is not suitable, the correction value should be re-adjusted until it is suitable.

(3) The references added in the revised manuscript are as below:
*1)Stuart, C. : An introduction to statistical modeling of extreme values, Springer, London, UK, 2011.*
*2)Maria, F.C., Marco, M.: Extreme-coastal-water-level estimation and projection: a comparison of statistical methods, Nat. Hazards Earth Syst. Sci., 22: 1109–1128, 2022.*

**3. I would suggest a discussion for the advantage and limitations of the proposed method in comparison with other methods.**
**Response:** Thanks for your suggestion. According to your suggestion, the section of Discussion has been modified in the revised manuscript. More detailed explanations of the advantage and limitations of the proposed method in this study have been presented in the section of Discussion.
(1) In the revised manuscript, the Discussion have been modified as below:
*4. Discussion*
*The warning water level is mainly used for storm surge prewarning, and it is crucial to decision-making and mitigation measure design. This study proposed a newly approved quantitative method for determining the four-color warning water level, which includes the calculation formula of the HWL at the typical return period, the classification method of the shore section based on its importance and coastal county unit,and the quantitative calculation formula of the correction value of the warning tide level corresponding to wave exposure degree, surge protection facility construction standard and the shore section importance level. Compared with the method used for calculating the one-single-value warning water level in the mid-1990s, the method of calculating the four-color warning water level used in this study is more reasonable, mainly in the following aspects: (1) It proposed the description of the warning water level classification corresponding to the four levels of marine disaster emergency response levels, and the determination results of the four-color warning tide level are more helpful for the storm surge prewarning; (2) The calculation of correction values has been improved, by replacing qualitative calculation method with quantitative calculation method, especially proposing the method of calculating the wave run-up which is an important decisive element for the correction values; (3) In the process of calculating the four-color warning water level, the verification of the approved results are strengthened, to determine whether the approved warning water level is suitable based on the statistical analysis of historical storm surge disasters and the corresponding tidal heights. Our results about the spatial distribution of four-color warning water level, have been preliminarily applied to storm surge disaster prevention and mitigation in coastal areas of China. Several studies focused on the storm surge prewarning application methods for the newly*

[revised manuscript text omitted]

| | | |
|---|---|---|
| | —Port throughput $\geq 3 \times 10^{11}$ kg/a;
—Construction investment $\geq 1.4 \times 10^9$ USD;
—Economic output of the protected area $\geq 7 \times 10^5$ USD/hm²/a;
—The cargo unloading capacity of the central fishing port $\geq 8 \times 10^7$ kg/a;
—Agricultural reclamation area $\geq 2 \times 10^3$ hm². | |
| Important | The shore section level is considered to be important if it meets one of the following conditions:
—Population density in the protected area = [400 persons/km², 1000 persons/km²);
—Port throughput = [$2 \times 10^{11}$ kg/a, $3 \times 10^{11}$ kg/a);
—Construction investment = [$0.7 \times 10^9$ USD, $1.4 \times 10^9$ USD);
—Economic output of the protected area = [$1.4 \times 10^5$ USD/hm²/a, $7 \times 10^5$ USD/hm²/a);
—The cargo unloading capacity of the first-class fishing port $\geq 4 \times 10^7$ kg/a;
—Agricultural reclamation area = [$6.67 \times 10^2$ hm², $2 \times 10^3$ hm²). | [−10,0) |
| Relatively important | The shore section level is considered to be important if it meets one of the following conditions:
—Population density in the protected area = [30 persons/km², 400 persons/km²);
—Port throughput = [$1 \times 10^{11}$ kg/a, $2 \times 10^{11}$ kg/a);
—Construction investment = [$0.14 \times 10^9$ USD, $0.7 \times 10^9$ USD);
—Economic output of the protected area = [$0.56 \times 10^5$ USD/hm²/a, $1.4 \times 10^5$ USD/hm²/a);
—The cargo unloading capacity of the second-class fishing port $\geq 2 \times 10^7$ kg/a;
—Agricultural reclamation area = [67 hm², 667 hm²). | [0,10) |
| Normal | The shore section level is considered to be normal if it meets one of the following conditions:
—Population density in the protected area < 30 persons/km²;
—Port throughput < $1 \times 10^{11}$ kg/a;
—Construction investment < $0.14 \times 10^9$ USD;
—Economic output of the protected area < $0.56 \times 10^5$ USD/hm²/a;
—The third-class fishing port can meet the berthing demand of local fishing boats;
—Agricultural reclamation area < 67 hm². | [10,20] |

Table note: The shore section level is categorized into four grades: particularly important, important, relatively important and normal. Each grade is mainly judged from 6 criteria, as long as one of the criteria is met, the shore section importance level can be considered to be this grade. The six criterion are population density, port throughput, construction investment, economic output, cargo unloading capacity and agricultural reclamation area. The value of $h_3$ can be taken as -20~20.

**5. The compass (the North Arrow) was missing in all figures.**

**Response:** Thanks for your suggestion. We have modified Fig. 1, Fig. 3, Fig. 4, Fig. 5 and Fig. 6 according to your advice, by adding the North Arrow to the above figures. The revised figures are as follows.

[Figure]

Fig. 1 The location of Zhifudao tide gauge station in Yantai City, Shandong Province, China

[Figure]

(a)                                    (b)

[Figure]

(c)                                          (d)

Fig. 3 Spatial distribution map of the four-color warning water level: a) Blue; b) Yellow; c)Orange; d)Red

[Figure]

Fig. 4 Spatial distribution map of the shore section importance level

[Figure]

(a)                               (b)

Fig.5 Spatial distribution map of $H_s$ and $H_d$: a) $H_s$; b) $H_d$

---

## Author Comment (AC2)

**Responses to Reference Report #2**

This paper proposes a quantitative method for determining the four-color warning water level, and the results show that proposed method could be easily adopted in various coastal areas. Especially, the study provides an insight into the spatial distribution of the four-color warning water level and its correction value along the coastlines of China. It can be helpful for storm-surge forecasting and prewarning. The paper is well structured and mostly easy to follow. However, there are few critical points that should be addressed in the manuscript as follow:

**Response**: We greatly appreciate your kind help in reviewing the manuscript and all constructive comments. We substantially revised the paper based on these comments.

1. Figure 4 gives the distribution map of the shore section importance level. It is very necessary to further clarify the distribution characteristics, for example, what is the proportion of the 259 shore sections corresponding to the different importance levels?

**Response:** Thanks for your suggestion. We have carried out a statistical analysis on the proportion of the 259 shore sections corresponding to the different importance levels. Among the 259 shore sections, 49.1% are particularly important shore sections, 32.4% are important shore sections, 13.1% are relatively important shore sections, and 5.4% are normal shore sections. The particularly important shore section accounts for the largest proportion(49.1%). More detailed explanations about the distribution characteristics of the shore section importance level have been presented in the revised manuscript as follows:

*Line 208-213:*

*The spatial distribution of shore section importance level were consistent with that of the warning water level. Among the 259 shore sections, the particularly important shore section accounted for the largest proportion(49.1%), while the other important grades shore sections accounted for 32.4%, 13.1% and 5.4% respectively. The shore section importance levels of Jiangsu, Zhejiang, Fujian, and Guangdong Provinces were higher than the other shore sections, and more than 90% of the particularly important shore sections were distributed in the coastal areas of the above provinces. This is because the coastal zones of these provinces with a high population density were the main areas of economic development on a country-wide scale, with this importance also being reflected in the high shore section importance level.*

2. The paper points out that the four-color warning water level corresponding to the four levels of marine disaster emergency response is more helpful for the storm surge prewarning. It is better to explain what is the marine disaster emergency response level and how the four warning water levels improve the marine disaster prevention capabilities compared to the previous system.

**Response:** Thanks for your suggestion. More detailed explanations about the storm surge disaster emergency response level and the role of the four-color warning water levels in marine disaster prevention have been presented in the revised manuscript.

*(1) Marine disaster emergency response is divided into four levels: Level I (particularly major disaster), Level II (major disaster), Level III (relatively major disaster), and Level IV (normal disaster). Marine disaster alerts are divided into four levels: red, orange, yellow, and blue,*

*corresponding to the highest to lowest warning levels, respectively. Correspondence between marine disaster emergency response and marine disaster alert level is shown in the table below.*

Table   Description of storm surge disaster emergency response level

| Marine disaster emergency response level | Description |
| --- | --- |
| I | Affected by tropical cyclones or extratropical weather systems, it is expected that the high tide level of one or more representative tide gauge stations in the affected area will reach the red warning tide level in the future, a red storm surge warning should be issued, and level I marine disaster emergency response level should be launched. |
| II | Affected by tropical cyclones or extratropical weather systems, it is expected that the high tide level of one or more representative tide gauge stations in the affected area will reach the orange warning tide level in the future, an orange storm surge warning should be issued, and level II marine disaster emergency response level should be launched. |
| III | Affected by tropical cyclones or extratropical weather systems, it is expected that the high tide level of one or more representative tide gauge stations in the affected area will reach the yellow warning tide level in the future, a yellow storm surge warning should be issued, and level III marine disaster emergency response level should be launched. |
| IV | Affected by tropical cyclones or extratropical weather systems, it is expected that the high tide level of one or more representative tide gauge stations in the affected area will reach the blue warning tide level in the future, a blue storm surge warning should be issued, and level IV marine disaster emergency response level should be launched. |

*(2) Compared with the single value format that characterized the warning water level as determined in the mid-1990s(the previous system), the four-color warning water level, corresponding to the four levels of marine disaster emergency response levels are more helpful for the storm surge prewarning. The advantages are mainly manifested in several aspects:1)In the north area of the Yangtze River Estuary and the South China Sea, the newly approved blue warning water level has changed the situation that most shore sections can rarely reach the high tide level of the blue storm surge warning level once in 5 to 15 years according to the original standard. It is conducive to improving the the warning service capability of the above-mentioned shore section, and have also eliminated the paralyzing thinking caused by the absence of super-warning tide levels for many years;2)In the coastal areas of Fujian Provice, the frequent issuance of blue storm surge warnings has been effectively avoided, so that it can better play the role of alerting tide levels;3)The newly approved red alert tide level can more truly reflect the actual defense capability of the approved shore section.*

3. The "Discussion" section should be further improved. For example, more detailed explanations of the advantage, limitation and future research could be presented.
**Response:** Thanks for your suggestion. According to your advice, the section of Discussion has been modified in the revised manuscript. More detailed explanations of the advantage, limitation and future research of this study have been presented in the section of Discussion.
(1) In the revised manuscript, the Discussion have been modified as below:
*4. Discussion*

*The warning water level is mainly used for storm surge prewarning, and it is crucial to decision-making and mitigation measure design. This study proposed a newly approved quantitative method for determining the four-color warning water level, which includes the calculation formula of the HWL at the typical return period, the classification method of the shore section based on its importance and coastal county unit,and the quantitative calculation formula of the correction value of the warning tide level corresponding to wave exposure degree, surge protection facility construction standard and the shore section importance level. Compared with the method used for calculating the one-single-value warning water level in the mid-1990s, the method of calculating the four-color warning water level used in this study is more reasonable, mainly in the following aspects: (1) It proposed the description of the warning water level classification corresponding to the four levels of marine disaster emergency response levels, and the determination results of the four-color warning tide level are more helpful for the storm surge prewarning; (2) The calculation of correction values has been improved, by replacing qualitative calculation method with quantitative calculation method, especially proposing the method of calculating the wave run-up which is an important decisive element for the correction values; (3) In the process of calculating the four-color warning water level, the verification of the approved results are strengthened, to determine whether the approved warning water level is suitable based on the statistical analysis of historical storm surge disasters and the corresponding tidal heights. Our results about the spatial distribution of four-color warning water level, have been preliminarily applied to storm surge disaster prevention and mitigation in coastal areas of China. Several studies focused on the storm surge prewarning application methods for the newly approved four-color warning water level, corresponding to a refined shore section (Fu et al., 2017). However, limited by the data availability, it is not considered that the influence of storm surge disaster loss factors on the calculation of warning water level. The Correlation between storm surge disaster losses and the highest tide water exceeding the warning water level has not been established.*

*The precision of the warning water level directly affects the accuracy of the storm surge prewarning results, thereby affecting the objectivity of emergency strategies and decision-making for storm surge disaster mitigation. With the rapid development of China's coastal society and economy, storm surge protection facilities, population density, and coastal development conditions have also been changing. Therefore, the warning water level needs to be updated according to the actual conditions of the coastal areas in time, When it is not compatible with the storm surge prevention and mitigation. At the same time, in order to meet the needs of the increasingly refined storm surge disaster prevention and mitigation plans, the scale of warning water level assessment should be changed from coastal counties to coastal towns and communities.*

*Several studies highlighted that global sea-level rise would continue accelerating in the 21st century as a consequence of climate change (Church and White, 2011; Hay et al., 2015). In fact, coastal flooding hazard has been increasing on a global scale in recent decades, a trend expected to continue as a result of climate change(Maria et al., 2022). In the past 40 years, sea level in the coastal China seas has increased significantly, with the rate of 3.4 mm/a, higher than the global average from 1993-2018(3.25mm/a)(Ministry of Natural Resources of China, 2021; IPCC,2021). In the IPCC Sixth Assessment Report, the latest monitoring and simulation results indicate that the current rate of Global mean sea level rise from 2006 to 2018 is accelerating (3.7mm/a) and will continue to rise in the future, showing an irreversible trend(Zhang et al., 2021; IPCC,2021).*

*Regional relative sea level rise is an important driving factor affecting extreme still water levels. The continuous rising sea level has led to an increase in extreme water levels in coastal areas of China(Qi et al., 2019), which can have an impact on the determination of warning water levels. Additionally, changes in storminess may have an important role in modifying the frequency and magnitude of water level extremes (Lowe et al., 2010; Woodworth et al., 2011). Future work about re-determining the warning water level should take these abovementioned issues into consideration.*

(2)The references added in the revised manuscript are as below:

*1) Maria, F.C., Marco, M.: Extreme-coastal-water-level estimation and projection: a comparison of statistical methods, Nat. Hazards Earth Syst. Sci., 22: 1109–1128, https://doi.org/10.5194/nhess-22-1109-2022, 2022.*

*2) Ministry of Natural Resources of China: 2021 China Sea Level Bulletin, Ministry of Natural Resources of China, Beijing, China, 2021. (in Chinese).*

*3) Zhang, T., Yu, Y.Q., Xiao, C.D.: Interpretation of IPCC AR6 report: monitoring and projections of global and regional sea level change, Climate Change Research, 17(6), 12–18, https://doi.org/10.12006/j.issn.1673-1719.2021.231, 2021.(in Chinese)*

*4) IPCC: Climate change 2021: The physical science basis. Contribution of working group I to the Sixth Assessment Report of the IPCC, Cambridge University Press, Cambridge, United Kingdom and New York, NY, USA, 2021.*

*5)Qi, Q.H., Cai, R.S., Yan, X.H.: Discussion on climate change and marine disaster risk governance in the coastal China seas, Marine Science Bulletin, 38(4), 361–367, https://doi.org/10.11840/j.issn.1001-6392.2019.04.001, 2019.*

4. U. S. or USA, China or PRC, please make it uniform in the entire manuscript.

**Response:** Thanks for your suggestion. We have checked the description of "USA" and "China", and make "USA" and "China" uniform in the entire manuscript.

5. Re-check the unit of Table 2 and 6. At the same time, it is recommended to move the unit "cm" of Table 6 from the table to the header position.

**Response:** Thanks for your suggestion. The unit of Table 2 and 6 have been changed to "a" in the revised manuscript. Based on your suggestion, we have also modified the positon of the unit "cm" in Table 5 as below:

Tab. 2 $H_s$ value corresponding to return period (unit: a)

| Corresponding water level return period of the actual defense capability of the shore section | Corresponding return period of $H_s$ |
|---|---|
| (0,50) | 2 |
| (50,100) | 3 |
| (100,200) | 4 |
| ≥200 | 5 |

Tab. 6 The high water levels (HWL) corresponding to return period at Zhifudao tide gauge station(unit: cm)

| Return period | 2a | 5a | 10a | 20a | 50a | 100a |
|---|---|---|---|---|---|---|

| HWL corresponding to return period | 184 | 209 | 225 | 240 | 260 | 275 |
| --- | --- | --- | --- | --- | --- | --- |

6. Figure 5 shows the spatial distribution of $H_s$ and $H_d$. Please re-check the name of Figure 5 and make sure the name corresponding to the content.

**Response:** Thanks for your suggestion. We have modified the name of Figure 5 in the revised manuscript as below:

*Fig.5 Spatial distribution map of $H_s$ and $H_d$: a) $H_s$; b) $H_d$*

---

## Author Comment (AC3)

**Responses to Reference Report #3**

The authors present new technological methods utilized for determining warning water levels, as well as the procedure and results of this determination in Zhifu District, Yantai City, Shandong Province, China. This study discovered the existing marine disaster prevention capacities of coastal areas by analyzing the spatial distribution patterns of warning water levels in 259 shore sections in China, and recommend changes for future warning water level evaluations based on their findings. Notably, this assessment can serve as a scientific reference for encouraging the redetermination of warning water levels in China's coastal areas, thereby improving their marine disaster prevention and protection capabilities. In summary, it is a topic of interest to the researchers in the related areas. This is a carefully done study and the findings are of much considerable interest. I recommend this manuscript to be published in NHESS. My detailed comments are as follows:

**Response**: We greatly appreciate your kind help in reviewing the manuscript and all constructive comments. We substantially revised the paper based on these comments.

1. The "Discussion" section needs to be improved. For example, sea level rise under the effects of global warming exhibits an accelerating trend and may potentially be irreversible. The impact of the ongoing sea level rise on the rise in severe water levels has to be covered in more detail and depth.

**Response:** Thanks for your suggestion. According to your advice, the section of Discussion has been modified in the revised manuscript, by adding some explanations of the impact of ongoing sea level rise on the rise in severe water levels.

*4.discussion*

*Line 282-288:*

*Several studies highlighted that global sea-level rise would continue accelerating in the 21st century as a consequence of climate change (Church and White, 2011; Hay et al., 2015). In fact, coastal flooding hazard has been increasing on a global scale in recent decades, a trend expected to continue as a result of climate change(Maria et al., 2022). In the past 40 years, sea level in the coastal China seas has increased significantly, with the rate of 3.4 mm/a, higher than the global average from 1993-2018(3.25mm/a)(Ministry of Natural Resources of China, 2021; IPCC,2021). In the IPCC Sixth Assessment Report, the latest monitoring and simulation results indicate that the current rate of Global mean sea level rise from 2006 to 2018 is accelerating (3.7mm/a) and will continue to rise in the future, showing an irreversible trend(Zhang et al., 2021; IPCC,2021). Regional relative sea level rise is an important driving factor affecting extreme still water levels. The continuous rising sea level has led to an increase in extreme water levels in coastal areas of China(Qi et al., 2019), which can have an impact on the determination of warning water levels. Additionally, changes in storminess may have an important role in modifying the frequency and magnitude of water level extremes (Lowe et al., 2010; Woodworth et al., 2011). Future work about re-determining the warning water level should take these abovementioned issues into consideration.*

The references added in the revised manuscript are as below:

*1) Maria, F.C., Marco, M.: Extreme-coastal-water-level estimation and projection: a comparison of statistical methods, Nat. Hazards Earth Syst. Sci., 22: 1109–1128, https://doi.org/10.5194/nhess-22-1109-2022, 2022.*

*2) Ministry of Natural Resources of China: 2021 China Sea Level Bulletin, Ministry of Natural Resources of China, Beijing, China, 2021. (in Chinese).*

*3) Zhang, T., Yu, Y.Q., Xiao, C.D.: Interpretation of IPCC AR6 report: monitoring and projections of global and regional sea level change, Climate Change Research, 17(6), 12–18, https://doi.org/10.12006/j.issn.1673-1719.2021.231, 2021.(in Chinese)*

*4) IPCC: Climate change 2021: The physical science basis. Contribution of working group I to the Sixth Assessment Report of the IPCC, Cambridge University Press, Cambridge, United Kingdom and New York, NY, USA, 2021.*

*5)Qi, Q.H., Cai, R.S., Yan, X.H.: Discussion on climate change and marine disaster risk governance in the coastal China seas, Marine Science Bulletin, 38(4), 361–367, https://doi.org/10.11840/j.issn.1001-6392.2019.04.001, 2019.*

2. In the "Discussion" section, more detailed explanations of the advantage, limitation of the technological method used in this study could be presented.

**Response:** Thanks for your suggestion. According to your advice, the section of Discussion has been modified in the revised manuscript. More detailed explanations of the advantage and limitation of the technological method used in this study have been presented in the section of Discussion.

In the revised manuscript, the Discussion have been modified as below:

*4. Discussion*

*The warning water level is mainly used for storm surge prewarning, and it is crucial to decision-making and mitigation measure design. This study proposed a newly approved quantitative method for determining the four-color warning water level, which includes the calculation formula of the HWL at the typical return period, the classification method of the shore section based on its importance and coastal county unit,and the quantitative calculation formula of the correction value of the warning tide level corresponding to wave exposure degree, surge protection facility construction standard and the shore section importance level. Compared with the method used for calculating the one-single-value warning water level in the mid-1990s, the method of calculating the four-color warning water level used in this study is more reasonable, mainly in the following aspects: (1) It proposed the description of the warning water level classification corresponding to the four levels of marine disaster emergency response levels, and the determination results of the four-color warning tide level are more helpful for the storm surge prewarning; (2) The calculation of correction values has been improved, by replacing qualitative calculation method with quantitative calculation method, especially proposing the method of calculating the wave run-up which is an important decisive element for the correction values; (3) In the process of calculating the four-color warning water level, the verification of the approved results are strengthened, to determine whether the approved warning water level is suitable based on the statistical analysis of historical storm surge disasters and the corresponding tidal heights.*

*Our results about the spatial distribution of four-color warning water level, have been preliminarily applied to storm surge disaster prevention and mitigation in coastal areas of China. Several studies focused on the storm surge prewarning application methods for the newly approved four-color warning water level, corresponding to a refined shore section (Fu et al., 2017). However, limited by the data availability, it is not considered that the influence of storm surge disaster loss factors on the calculation of warning water level. The Correlation between storm surge disaster losses and the highest tide water exceeding the warning water level has not been established.*

*The precision of the warning water level directly affects the accuracy of the storm surge prewarning results, thereby affecting the objectivity of emergency strategies and decision-making for storm surge disaster mitigation. With the rapid development of China's coastal society and economy, storm surge protection facilities, population density, and coastal development conditions have also been changing. Therefore, the warning water level needs to be updated according to the actual conditions of the coastal areas in time, When it is not compatible with the storm surge prevention and mitigation. At the same time, in order to meet the needs of the increasingly refined storm surge disaster prevention and mitigation plans, the scale of warning water level assessment should be changed from coastal counties to coastal towns and communities.*

*Several studies highlighted that global sea-level rise would continue accelerating in the 21st century as a consequence of climate change (Church and White, 2011; Hay et al., 2015). In fact, coastal flooding hazard has been increasing on a global scale in recent decades, a trend expected to continue as a result of climate change (Maria et al., 2022). In the past 40 years, sea level in the coastal China seas has increased significantly, with the rate of 3.4 mm/a, higher than the global average from 1993-2018(3.25mm/a)(Ministry of Natural Resources of China, 2021; IPCC,2021). In the IPCC Sixth Assessment Report, the latest monitoring and simulation results indicate that the current rate of Global mean sea level rise from 2006 to 2018 is accelerating (3.7mm/a) and will continue to rise in the future, showing an irreversible trend (Zhang et al., 2021; IPCC,2021). Regional relative sea level rise is an important driving factor affecting extreme still water levels. The continuous rising sea level has led to an increase in extreme water levels in coastal areas of China(Qi et al., 2019), which can have an impact on the determination of warning water levels. Additionally, changes in storminess may have an important role in modifying the frequency and magnitude of water level extremes (Lowe et al., 2010; Woodworth et al., 2011). Future work about re-determining the warning water level should take these abovementioned issues into consideration.*